# High Incidence of Atopic Dermatitis among Children Whose Fathers Work in Primary Industry: The Japan Environment and Children’s Study (JECS)

**DOI:** 10.3390/ijerph19031761

**Published:** 2022-02-03

**Authors:** Hiroshi Yokomichi, Mie Mochizuki, Reiji Kojima, Sayaka Horiuchi, Tadao Ooka, Yuka Akiyama, Kunio Miyake, Megumi Kushima, Sanae Otawa, Ryoji Shinohara, Zentaro Yamagata

**Affiliations:** 1Department of Health Sciences, University of Yamanashi, Chuo 4093898, Yamanashi, Japan; kojimar@yamanashi.ac.jp (R.K.); tohoka@yamanashi.ac.jp (T.O.); yukaa@yamanashi.ac.jp (Y.A.); kmiyake@yamanashi.ac.jp (K.M.); zenymgt@yamanasahi.ac.jp (Z.Y.); 2Department of Paediatrics, University of Yamanashi, Chuo 4093898, Yamanashi, Japan; mmie@sweet.ocn.ne.jp; 3Centre for Birth Cohort Studies, University of Yamanashi, Chuo 4093898, Yamanashi, Japan; hsayaka@yamanashi.ac.jp (S.H.); kumegumi@yamanashi.ac.jp (M.K.); osanae@yamanashi.ac.jp (S.O.); rshinohara@yamanashi.ac.jp (R.S.)

**Keywords:** hygiene hypothesis, atopic dermatitis, agriculture, children, livestock, forestry industry, endotoxin, urban living, country living, primary industry

## Abstract

The hygiene hypothesis assumes a low incidence of allergic diseases in families engaging in farming work. However, a few studies have indicated rural life as a potential risk factor for atopic dermatitis (AD). Using a large Japanese birth cohort dataset, we calculated the accumulated incidence of AD in children aged 6 months and 1, 2, and 3 years by family business and the hazard ratio. We adjusted for confounding factors. The father’s job was considered the family business. We analysed data on 41,469 father–child pairs at 6 months of age, 40,067 pairs at 1 year, 38,286 pairs at 2 years, and 36,570 pairs at 3 years. We found the highest accumulated incidence of AD among children with fathers engaged in primary industry, with 2.5% at the age of 6 months, 6.6% at 1 year, 12.0% at 2 years, and 15.4% at 3 years. Among primary industry occupations, forestry was associated with the highest incidence of AD across these ages. The hazard ratio of AD was also highest for children whose family business was primary industry. In conclusion, the highest incidence and hazard ratio of AD were observed among children whose fathers worked in primary industry.

## 1. Introduction

Atopic dermatitis (AD) is more prevalent in industrialised countries than in developing nations [1]. Approximately 10–20% of Japanese children are affected by this condition [2]. The hygiene hypothesis posits that a small family size and cleanliness may lead to a high incidence of pollen allergies and eczema [3]. A study from Sweden showed that children who grew up with farm animals and those who ate fish had a lower incidence of allergic rhinitis [4]. This evidence suggests that, in industrialised nations, children in farming households should have a relatively low prevalence of AD.

A systematic review of international studies showed a lower prevalence of AD in rural regions than in urban regions [5]. In a British birth cohort study, a farm residence was associated with a 10% decrease in the incidence of AD in children, although this relationship was nonsignificant [6]. A Finnish population study also reported a nonsignificant association between a farm residence and a lower incidence of AD [7]. There may be a developing consensus that a rural residence is a preventive environmental factor for AD [8].

However, in the modern era, when many countries have become highly industrialised, the hygiene hypothesis may need to be modified [9]. A previous study of primary schoolchildren in Japan showed a higher prevalence of AD in several rural regions than in urban regions [10]. This evidence suggests the need to microscopically investigate children’s environmental risk factors for AD. Life in rural regions needs to be distinguished by whether it involves exposure to animals and plants. In industrialised nations, almost all citizens live in much cleaner environments than people have previously experienced. Environmental exposure to natural allergens may influence the risk of allergic disease among people who live in regions with undisturbed nature. Children whose parents’ business or occupation is primary industry tend to live near natural plants and animals. In this study, we investigated the incidence and prevalence of AD according to children’s family business in a large Japanese birth cohort [11,12,13,14,15]. We found that children whose fathers worked in primary industry had the highest incidence and prevalence of AD.

## 2. Materials and Methods

The Japan Environment and Children’s Study (JECS) project details have been published in [11]. The JECS comprises a cohort of 104,062 children born from 2011 to 2014 in 15 Regional Centres covering 19 prefectures across Japan [16]. Exposure to environmental factors was assessed using chemical analyses of biospecimens, household environment measurements, and computational simulations based on monitoring data and questionnaires. One of the JECS’s prioritized foci was immune system disorders (allergic diseases) [11].

We used data on children’s history of AD and the father’s job (as a measure of the family business). In the questionnaire, the response options for the father’s job were management, qualified professional, engineering, teaching or art, desk work, sales, service, security service, primary industry, production, transportation driving, and construction. Primary industry comprised agriculture, forestry, and fisheries. A parent or caregiver responded to a questionnaire item on whether the child had been diagnosed with AD by a physician at the ages of 6 months, 1 year, 2 years, and 3 years. The family business was recorded during pregnancy, and experience of the diagnosis of AD was recorded at the corresponding ages.

We calculated the incidence of AD as the accumulated experience of the diagnosis of AD until the oldest age studied. We constructed a graph of the crude accumulated incidence of AD by the family business and the child’s age. We determined the prevalence of AD as the proportion of children diagnosed with AD at the ages mentioned above. The accumulated incidence was measured in a cohort design, and the prevalence was measured in a cross-sectional design.

We used the fixed-effect model to calculate the least square means of the accumulated incidence and prevalence of AD [17] at each time point according to the family business. In this model, we adjusted for the parents’ history of AD [18], the parents’ history of current smoking [19], the parents’ educational backgrounds [20], family income [21], and whether the family had a dog or a cat [22].

In the cohort study, the accumulated incidence of AD was also used for the outcome of the time-to-event analysis. The incidence of AD was regressed by the Cox proportional hazard model. In the cross-sectional study, the prevalence at each time point was used for the outcome. The prevalence of AD was regressed by the logistic model. We calculated hazard ratios (HRs) of the incidence of AD and odds ratios (ORs) of the prevalence of AD between primary industry family businesses and other family businesses. In the calculation of HRs, children who were lost to follow-up were censored. The HRs and ORs were adjusted for the above-mentioned covariates. We used the jecs-ta-20190930-qsn JECS dataset. We conducted all statistical analyses using SAS, Version 9.4 (SAS Institute, Cary, NC, USA). Two-sided *p*-values < 0.05 were considered to indicate a statistically significant difference.

## 3. Results

Figure 1 shows the crude percentage of the accumulated incidence of AD in children according to the family business. Table 1 shows the adjusted percentage of the accumulated incidence of AD in children according to the family business. Children whose fathers worked in agriculture, forestry, or fisheries (primary industrial business) had the highest incidence of AD (2.5% at 6 months of age, 6.6% at 1 year, 12.0% at 2 years, and 15.4% at 3 years). Among these three primary industries, agriculture showed a high incidence of AD (2.4% at 6 months of age, 5.9% at 1 year, 10.8% at 2 years, and 13.3% at 3 years), and forestry showed the highest incidence in the same age groups (3.3%, 6.7%, 15.8%, and 25.0%, respectively). Table 2 shows the adjusted prevalence of AD by age. Children whose fathers worked in agriculture, forestry, or fisheries had the highest prevalence of AD (2.5% at 6 months of age, 6.0% at 1 year, 6.4% at 2 years, and 8.2% at 3 years).

Table 3 shows the adjusted HRs of the incidence of AD for other family businesses vs. primary industry businesses. The HRs of the other businesses were less than one at the ages of 6 months to 3 years, which indicated that primary industry businesses as the reference category had the highest incidence of AD. In the subcategory of primary industry businesses, forestry showed an HR of 1.73, which was not significant compared to agriculture. This finding suggested that children whose family business was forestry had the highest incidence of AD.

Table 4 shows the adjusted ORs of the prevalence of AD for other family businesses vs. primary industry businesses. The ORs of the other businesses were less than one compared with primary industry businesses, which suggested that children whose family business was one other than primary industry had a lower prevalence of AD. In the subcategory of primary industry businesses, forestry showed an OR of 2.33, which was significant at the age of 3 years. This finding suggested that children with the family business of forestry had the highest prevalence of AD.

## 4. Discussion

In contrast to our expectations, the incidence and prevalence of AD at the ages of 6 months to 3 years were higher in children with fathers working in primary industry than in children with fathers working in other areas. Among the primary industry jobs, forestry was associated with the highest incidence of AD.

These results are inconsistent with the hygiene hypothesis and with previously reported findings suggesting that a higher socioeconomic status, higher levels of family education, and urban residence are associated with a higher risk of AD [23]. Another recent study also suggested that greater environmental exposure to farming life was associated with a low incidence of AD [24]. One explanation for the contradiction may be that only a subset of families engaged in primary industry in our data can be considered to live a farming lifestyle. This situation in Japan, which is a highly industrialised country, may have reduced the difference in the incidence and prevalence of AD between occupations. In regions rich in natural resources where many people work in primary industry, children may be exposed to a relatively high level of environmental allergens.

Recently, pollen allergies were designated a national disease in Japan [25]. Pollen allergens, such as alder, Japanese cedar [26], Japanese cypress, rice, and ragweed, grow wild in Japan and are grown commercially in rural regions. A Japanese study in 2006 reported the following prevalence levels among schoolchildren: 27.4% for allergic rhinitis, 25.2% for allergic conjunctivitis, and 5.6% for AD [25]. Exposure to pollen significantly worsens AD [27]. The reason for our finding of a high incidence of AD associated with forestry may be that children who reside in mountainous regions are exposed to a large amount of pollen. High environmental exposure to pollen may increase the risk of AD.

Confounding bias may not be sufficiently reduced by using multivariable analysis. The smoking rate is high in Japanese men working in fishery, agriculture, forestry, and mining [28]. Because parental smoking increases the incidence of AD [29], residual bias of paternal smoking may have increased the incidence of AD in our study.

We found that having a dog or a cat was associated with a higher incidence of AD in children (data not shown). This finding suggests that, in the modern era, living with animals does not directly correspond to a reduced allergic disease risk. However, in a German study, having a dog or a cat was shown to be protective against AD for children aged 1 or 2 years [30], and a previous Japanese study showed that cat ownership was a preventive factor, but dog ownership was not [31]. A meta-analysis also supported the association between pet ownership and a reduced incidence of AD, but showed no association for cat ownership [22]. Because several studies included in the meta-analysis found a positive association or no association between pet ownership and the risk of AD, there is no clear consensus regarding pet ownership as a preventive factor for AD. If such a protective effect exists, it may occur through changes in intestinal colonisation patterns during infancy [32] and through effects on the maturation of the immune system [33].

Another confounder in an infant’s life may be the feeding technique used. Epidemiologically, breast feeding may provide a reduced risk of child allergic diseases [34], whereas formula milk feeding may result in an increased risk [35]. However, the JECS data suggested an association between early formula milk intake and a reduced risk of cow’s milk allergy [36]. Additionally, drinking cow’s milk might be associated with a higher risk of developing allergies [37,38]. If the family business is related to milk consumption, the present results may have been affected by the children’s amount of milk consumption.

In this study, the number of participants with forestry as the family business may have been too small to inform meaningful conclusions about this group. The number of participants with agriculture (approximately 400 participants) and fisheries (approximately 150 participants) as the family business was sufficient for estimating the incidence and prevalence of AD. However, the small number of participants with a family business of forestry (approximately 100 participants) may have reduced the precision of the estimation. The relatively small number of participants with forestry as their family business may be explained by the decline in the number of forestry workers in Japan in recent years.

Farm exposure protects people from developing asthma [39]. A reduced exposure to microorganisms and increased exposure to allergens in modern indoor life might increase the prevalence of rhinitis [40]. A protective effect from AD was found with early day care, endotoxin, and animal exposure [41]. However, recent scientific advances suggested that the hygiene hypothesis is complex [42]. In contrast to this hypothesis, which posits that children with parents working in primary industry have a low risk of allergic disease, our study identified primary industry as a risk factor for AD.

Our finding of a higher risk of AD in children with fathers in primary industry may be indicative of a higher risk in children living in rural regions. This finding may be explained by the household socioeconomic status [21]. The prevalence of AD is associated with a low socioeconomic status [2]. Because incomes in forestry are not high in Japan, a high incidence of AD in children with fathers who work in forestry (Table 1) may be observed. A low income may result in parents not having sufficient time or money to care for their children’s skin. Although the mechanism for this association is unknown, preventive behaviours might need to be emphasised in families engaged in primary industry.

Our study has several limitations. First, we could not collect data on the family business of livestock farming. The agriculture category was a mixture of crop farming and livestock farming. Second, the development of AD was reported by caregivers on the basis of a physician’s diagnosis. Therefore, caregivers could have exhibited recall bias. Third, our results were limited by the detection of *Staphylococcus aureus* in skin not being examined as a potential confounder. Fourth, the physicians who diagnosed the children in this study included AD specialists and nonspecialists. Therefore, the physicians may have underdiagnosed AD in infancy because this diagnosis can cause stigma for children and caregivers [43]. Fifth, the father’s occupation may have changed between pregnancy and the child reaching 3 years of age. The father may also have been replaced by a new partner, and, therefore, the family business may have changed throughout the studied ages. This possibility could have decreased the differences in the incidence or prevalence of AD between occupational categories. Sixth, although the incidence and the prevalence of AD in primary industry was higher than that in the other occupations, there were almost no significant differences in the comparisons (Table 1; Table 2). We hypothesised that this was due to a low statistical power. Finally, recent literature suggested that epigenetics modify the exposure and outcome relationship [44]. We were not able to consider this issue in our study.

## 5. Conclusions

In Japan, AD is more prevalent in children whose fathers work in primary industry. This finding suggests an increased risk of AD in rural regions.

## Figures and Tables

**Figure 1 ijerph-19-01761-f001:**
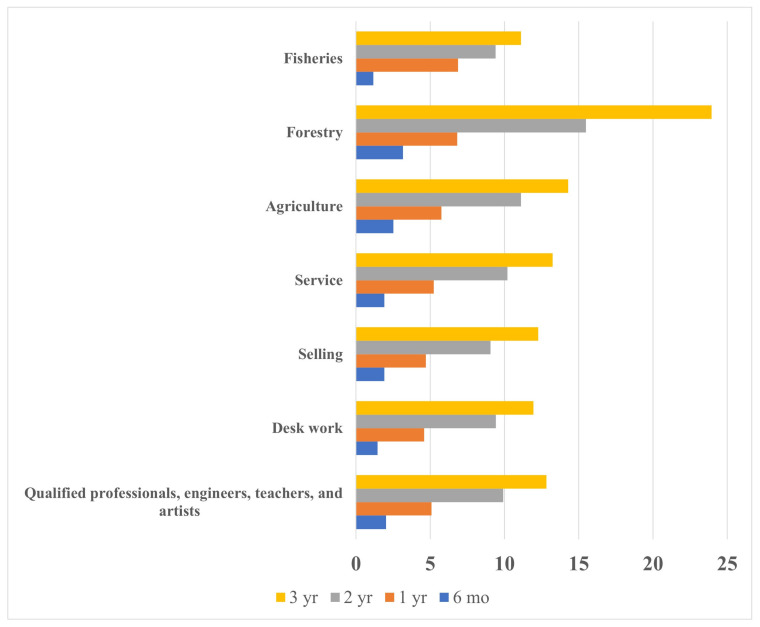
The crude accumulated incidence of atopic dermatitis in preschool children by the family business and the child’s age.

**Table 1 ijerph-19-01761-t001:** The accumulated incidence of atopic dermatitis in preschool children by the family business and the child’s age.

	Child’s Age			
Paternal Business	6 Months	1 Year	2 Years	3 Years
Category, no. (adjusted percentage)				
Management	38/1831 (2.0)	92/1785 (5.1)	175/1695 (10.4)	208/1625 (12.5)
Qualified professionals, engineers, teachers, and artists	270/13,322 (1.9)	657/12,934 (5.0)	1219/12,310 (9.8)	1517/11,838 (12.9)
Desk work	61/4186 (1.4)	187/4085 (4.6)	366/3889 (9.6)	449/3759 (12.7)
Selling	89/4671 (1.9)	212/4503 (4.7)	387/4269 (9.0)	500/4076 (12.2)
Service	88/4607 (1.9)	231/4409 (5.3)	422/4141 (10.1)	525/3966 (13.1)
Security service	35/1862 (1.8)	93/1792 (5.2)	158/1722 (9.2)	198/1668 (12.0)
Agriculture, forestry, and fisheries	16/703 (2.5)	41/666 (6.6)	71/629 (12.0)	86/586 (15.4)
Production	96/5726 (1.8)	246/5535 (4.5)	453/5251 (8.6) *	574/5014 (11.3) *
Transportation driver	25/1767 (1.6)	79/1701 (4.8)	153/1596 (9.6)	193/1525 (12.6)
Construction	41/2794 (1.7)	117/2657 (4.5)	235/2462 (9.5)	295/2335 (12.5)
Total, no. (crude calculation)	759/41,469 (1.8)	1955/40,067 (4.9)	3639/37,964 (9.6)	4545/36,392 (12.5)
Subcategory, no. (adjusted percentage)				
Agriculture	11/437 (2.4)	24/418 (5.9)	44/396 (10.8)	53/371 (13.3)
Forestry	3/95 (3.3)	6/88 (6.7)	13/84 (15.8)	17/71 (25.0) *
Fisheries	2/171 (1.3)	11/160 (6.5)	14/149 (9.8)	16/144 (12.6)

Percentages were adjusted using the least square mean method for the parents’ history of atopic dermatitis and current smoking, the parents’ educational background, family income, and the family having a dog or a cat. * *p* < 0.05 compared to the percentage for the agriculture, forestry, and fisheries category or compared to the percentage for the agriculture subcategory.

**Table 2 ijerph-19-01761-t002:** The prevalence of atopic dermatitis in preschool children by the family business and the child’s age.

	Child’s Age			
Family Business	6 Months	1 Year	2 Years	3 Years
Category, no. (adjusted percentage)				
Management	38/1831 (2.0)	78/1785 (4.4)	98/1725 (5.7)	110/1668 (6.7)
Qualified professionals, engineers, teachers, and artists	270/13,322 (1.9)	557/12,923 (4.2)	705/12,459 (5.6)	879/12,073 (7.3)
Desk work	61/4186 (1.4)	163/4084 (4.0)	226/3938 (5.9)	258/3817 (7.0)
Selling	89/4671 (1.9)	176/4499 (3.9)	224/4331 (5.1)	286/4158 (6.8)
Service	88/4607 (1.9)	192/4407 (4.4)	269/4211 (6.4)	308/4068 (7.5)
Security service	35/1862 (1.8)	79/1791 (4.4)	87/1749 (5.0) *	110/1697 (6.6)
Agriculture, forestry, and fisheries	16/703 (2.5)	37/665 (6.0)	38/646 (6.4)	47/609 (8.2)
Production	96/5726 (1.8)	222/5534 (4.0)	267/5326 (5.0)	340/5119 (6.6)
Transportation driver	25/1767 (1.6)	67/1701 (4.0)	92/1618 (5.7)	117/1551 (7.5)
Construction	41/2794 (1.7)	100/2656 (3.8)	138/2533 (5.4)	174/2423 (7.1)
Total, no. (crude calculation)	759/41,469 (1.8)	1671/40,045 (4.2)	2144/38,536 (5.6)	2629/37,183 (7.1)
Subcategory, no. (adjusted percentage)				
Agriculture	11/437 (2.4)	21/417 (5.1)	29/407 (6.9)	29/386 (6.6)
Forestry	3/95 (3.3)	5/88 (5.7)	3/86 (3.8)	9/73 (13.4)
Fisheries	2/171 (1.3)	11/160 (6.6)	6/153 (4.2)	9/150 (7.5)

Percentages were adjusted using the least square mean method for the parents’ history of atopic dermatitis and current smoking, the parents’ educational background, family income, and the family having a dog or a cat. * *p* < 0.05 compared to the percentage for the agriculture, forestry, and fisheries category or compared to the percentage for the agriculture subcategory.

**Table 3 ijerph-19-01761-t003:** The hazard ratio ^†^ of incident atopic dermatitis in preschool children for the family business.

Family Business	Hazard Ratio (95% CI)
Category	
Management	0.84 (0.65, 1.08)
Qualified professionals, engineers, teachers, and artists	0.83 (0.66, 1.03)
Desk work	0.79 (0.62, 0.99)
Selling	0.78 (0.62, 0.99)
Service	0.84 (0.67, 1.05)
Security service	0.78 (0.61, 1.01)
Agriculture, forestry, and fisheries	Reference
Production	0.74 (0.59, 0.93)
Transportation driver	0.81 (0.63, 1.05)
Construction	0.80 (0.63, 1.02)
Subcategory	
Agriculture	Reference
Forestry	1.73 (0.98, 3.05)
Fisheries	0.93 (0.52, 1.58)

^†^ Hazard ratios were adjusted for the parents’ history of atopic dermatitis and current smoking, the parents’ educational background, family income, and the family having a dog or a cat.

**Table 4 ijerph-19-01761-t004:** The odds ^†^ ratio of prevalent atopic dermatitis in preschool children by the family business and the child’s age.

	Child’s Age			
Family Business	6 Months	1 Year	2 Years	3 Years
Category				
Management	0.78	0.70	0.89	0.79
Qualified professionals, engineers, teachers, and artists	0.74	0.68 *	0.86	0.87
Desk work	0.54 *	0.64 *	0.9	0.83
Selling	0.72	0.62 *	0.79	0.81
Service	0.76	0.70	0.99	0.90
Security service	0.71	0.70	0.76	0.77
Agriculture, forestry, and fisheries	Reference	Reference	Reference	Reference
Production	0.68	0.64	0.75	0.78
Transportation driver	0.59	0.64	0.87	0.90
Construction	0.63	0.61 *	0.83	0.85
Subcategory				
Agriculture	Reference	Reference	Reference	Reference
Forestry	1.40	1.19	0.52	2.33 *
Fisheries	0.50	1.32	0.59	1.13

^†^ Odds ratios were adjusted for the parents’ history of atopic dermatitis and current smoking, the parents’ educational background, family income, and the family having a dog or a cat. * *p* < 0.05 compared to the percentage for the agriculture, forestry, and fisheries category or compared to the percentage for the agriculture subcategory.

## Data Availability

Data are available on reasonable request. Data are unsuitable for public deposition because of ethical restrictions and the legal framework of Japan. The Act on the Protection of Personal Information (Act No. 57 of 30 May 2003, amendment on 9 September 2015) prohibits public deposition of the data containing personal information. Ethical Guidelines for Medical and Health Research Involving Human Subjects enforced by the Japan Ministry of Education, Culture, Sports, Science and Technology and the Ministry of Health, Labour and Welfare also restrict the open sharing of epidemiologic data. All inquiries about access to data should be sent to: jecs-en@nies.go.jp. The person responsible for handling enquiries sent to this email address is Shoji F. Nakayama, JECS Programme Office, National Institute for Environmental Studies.

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
