# Peer review of "High Incidence of Atopic Dermatitis among Children Whose Fathers Work in Primary Industry: The Japan Environment and Children’s Study (JECS)"

_ijerph, 2022, doi:10.3390/ijerph19031761_

Round 1
Reviewer 1 Report
The reviewer appreciate the authors for adequately responding to earlier comments.
The manuscript is significantly improved and now reflect measures of association: ORs and HRs.
The reviewer still has some concerns with reporting these two associations. How was the HR ratio determined and calculated.
HR is reported in epidemiological studies when there is a time-to-event analysis in mind. That is when you are interested in knowing how long it takes for a particular event/outcome to occur.
In this case of the current study, what is the time-to-event outcome? I will assume it is AD. But was time taken into account in the analysis. I am struggling to understand if this was a follow-up study with censoring of participants who had the disease of interest or lost to follow-up until the end of the study.
In the method section, the authors stated: "A parent or caregiver responded to a questionnaire item on whether the child had been diagnosed with AD by a physician at the ages of 6 months, 1 year, 2 years, and 3 years."
If a child was diagnosed as AD case in the 6 months and the parent responded "Yes" at 6 month, 1 yr, 2yr and 3 yr; will this child be counted over this time period in the author's HR analysis?
The authors reported both HR and ORs. Please, can the author clarified the rationale for each of these two measures of association? In what situation was HR used and as well as OR.
This will allow readers to put the results of the study in perspective.
Author Response
Responses to comments from reviewer #1
- The reviewer appreciates the authors for adequately responding to earlier comments.
The manuscript is significantly improved and now reflect measures of association: ORs and HRs.
The reviewer still has some concerns with reporting these two associations. How was the HR ratio determined and calculated?
HR is reported in epidemiological studies when there is a time-to-event analysis in mind. That is when you are interested in knowing how long it takes for a particular event/outcome to occur.
In this case of the current study, what is the time-to-event outcome? I will assume it is AD. But was time taken into account in the analysis. I am struggling to understand if this was a follow-up study with censoring of participants who had the disease of interest or lost to follow-up until the end of the study.
In the method section, the authors stated: "A parent or caregiver responded to a questionnaire item on whether the child had been diagnosed with AD by a physician at the ages of 6 months, 1 year, 2 years, and 3 years."
If a child was diagnosed as AD case in the 6 months and the parent responded "Yes" at 6 month, 1 yr, 2yr and 3 yr; will this child be counted over this time period in the author's HR analysis?
The authors reported both HR and ORs. Please, can the author clarify the rationale for each of these two measures of association? In what situation was HR used and as well as OR.
This will allow readers to put the results of the study in perspective.
We appreciate the reviewer’s comments on how to improve our manuscript. The tables for this study were constructed using a cross-sectional study approach for the prevalence of AD (Tables 2 and 4) and using a cohort study approach for the incidence of AD (Tables 1 and 3). In the cohort study, the accumulated incidence of AD was used for the outcome of the time-to-event analysis. The incidence of AD was regressed by the Cox hazard model. HRs were used to describe the association between the incidence of AD and the family business. In the cross-sectional study, the prevalence at each time point was used for the outcome. ORs were used to describe the association between the prevalence of AD and the family business. We have clarified these measures of associations, the study designs, and the rationale behind our choices in the revised manuscript as follows.
Line 81: We calculated the incidence of AD as the accumulated experience of the diagnosis of AD until the oldest age mentioned above. We constructed a graph of the crude accu-mulated incidence of AD by the family business and the child’s age. We determined the prevalence of AD as the proportion of diagnosis of AD at the ages mentioned above. The accumulated incidence was measured in a cohort design, and the prevalence was measured in a cross-sectional design.
Line 92: The accumulated incidence was also used for the calculation of hazard ratios (HRs) by the Cox proportional hazard model. The prevalence was used for calculation of odds ratios (ORs). We calculated HRs of the incidence of AD and ORs of the prevalence of AD between primary industry family businesses and other family businesses. In the calculation of HRs, children who were lost to follow-up were censored.

Reviewer 2 Report
Thanks for showing me a good thesis. It would be nice to add a display format. It would be nice if you could show a circular statistic graph or a bar-shaped statistic graph. Thanks for adding it.
Author Response
Responses to comments from reviewer #2
- Thanks for showing me a good thesis. It would be nice to add a display format. It would be nice if you could show a circular statistic graph or a bar-shaped statistic graph. Thanks for adding it.
We thank the reviewer for the suggestion. We have added a bar graph illustrating the crude accumulated incidence of AD for the representative family business. We hope that this figure will help improve our manuscript.
Figure 1. Crude accumulated incidence of atopic dermatitis in preschool children by the family business and the child’s age.

Reviewer 3 Report
My comments have been addressed well. Thank you. I have no further reservations.
Author Response
Responses to comments from reviewer #3
- My comments have been addressed well. Thank you. I have no further reservations.
We thank the reviewer for suggestions that have improved our manuscript. We hope that our study will add evidence to the recent situation of AD.

Round 2
Reviewer 1 Report
Please, can the authors summarize this methodology somewhere in the method section?
" In the cohort study, the accumulated incidence of AD was used for the outcome of the time-to-event analysis. The incidence of AD was regressed by the Cox hazard model. HRs were used to describe the association between the incidence of AD and the family business. In the cross-sectional study, the prevalence at each time point was used for the outcome. ORs were used to describe the association between the prevalence of AD and the family business. We have clarified these measures of associations, the study designs, and the rationale behind our choices in the revised manuscript as follows."
This will give readers a clear indications of what what done
Author Response
Please, can the authors summarize this methodology somewhere in the method section?
" In the cohort study, the accumulated incidence of AD was used for the outcome of the time-to-event analysis. The incidence of AD was regressed by the Cox hazard model. HRs were used to describe the association between the incidence of AD and the family business. In the cross-sectional study, the prevalence at each time point was used for the outcome. ORs were used to describe the association between the prevalence of AD and the family business. We have clarified these measures of associations, the study designs, and the rationale behind our choices in the revised manuscript as follows."
This will give readers a clear indication of what has been done.
We thank the reviewer for the helpful comment. We have added sentences and have revised the description. We hope that the methodology has been sufficiently described.
Line 81: We calculated the incidence of AD as the accumulated experience of the diagnosis of AD until the oldest age mentioned above. We constructed a graph of the crude accumulated incidence of AD by the family business and the child’s age. We determined the prevalence of AD as the proportion of diagnosis of AD at the ages mentioned above. The accumulated incidence was measured in a cohort design, and the prevalence was measured in a cross-sectional design.
Line 92: In the cohort study, the accumulated incidence of AD was used for the outcome of the time-to-event analysis. The incidence of AD was regressed by the Cox proportional hazard model. In the cross-sectional study, the prevalence at each time point was used for the outcome. The prevalence of AD was regressed by the logistic model. We calculated hazard ratios (HRs) of the incidence of AD and odds ratios (ORs) of the prevalence of AD between primary industry family businesses and other family businesses. In the calculation of HRs, children who were lost to follow-up were censored. The HRs and ORs were adjusted for the above-mentioned covariates.

This manuscript is a resubmission of an earlier submission. The following is a list of the peer review reports and author responses from that submission.
Round 1
Reviewer 1 Report
Comments to authors
The study investigated the incidence of AD among children (at four life stages: 6 months, 1 year, 2 years and 3 years) whose fathers worked in different occupational categories with a specific focus on primary industry. Primary industry as defined by the authors included a combination of three farming occupational categories: Agriculture, Forestry and Fisheries. The authors investigated Primary Industry as a group and as individual subcategories and related this to the incidence of AD. The result of the study showed that the highest incidence and prevalence of AD occurred among children whose fathers worked in primary industry; suggesting an increased risk of AD among this population group compared to other occupational categories. While the intents of the authors were clear, the study had some serious flaws and a lot of things were unclear in the study.
First of all, this is a very difficult paper to read and comprehend. Also, it appears that the appropriate analyses were not used to present the results of the study.
From the introduction to the conclusion session, it was difficult to follow the idea and objective of the study clearly. The concept of incidence and prevalence are used interchangeably and none of the analysis reflected a degree of association. The methodology, while moderate, lacked basic association models in the analysis section. All of the analyses were based on simple descriptive analyses that do not reflect measures of association. So also are the results and discussion sessions. Specific comments are as follow:
SPRCIFIC COMMENTS
MAJOR COMMENTS
The major flaw in this study is the lacked of measures of association in all of the results presented.
On page 2; Lines 53–55; the authors mentioned: “…, we investigated the association… Japanese birth cohort”. However, the analysis presented were basic descriptive analysis that did not show any measure of associations. In a study as this, a measure of association either through odds ratio, hazard ratio etc will suffice and should be presented in addition to descriptive analysis shown by the authors.
The authors mentioned that they adjusted for confounders in the footnotes of Tables 1 and 2. How were confounders adjusted for in the results if there were no multivariate models to present measures of associations. I am struggling to understand how confounders were assessed and adjusted for in the results.
Furthermore, how were the authors able to account for occupational change among the fathers? For example, if a father was in Primary Industry at 6 months of assessment and at 2 years of assessment had changed occupation. How was crossover or change of occupation accounted in the relationship between occupational categories and AD?
AD was examined as outcome in this study based on questionnaire response. However, the authors mentioned that they assessed incidence of AD. How was AD assessed as incidence if they parents were only responding to questionnaire to indicate if their children had AD at the time of enquiry. To assess incidence of a disease, the exposure must precede the occurrence of the disease. Can the authors clearly explain how incidence was measured in this study. In epidemiological terms, it appears the authors are mainly assessing prevalence using a cross sectional study approach.
Were the child-father paired at 6 months also the same child-father pair at 1, 2 and 3 years; albeit a subset of the population at 6 month? This is not clear in the study. If the child-father pair at 6 months were the same population followed up at 1, 2, and 3 years to assess incidence of AD, then the measure of association such as hazard ratio and the person-year should be reported to reflect incidence risk.
Discussion
Page 2: Lines 121–124: If parental smoking was a risk factor for AD, I am curious to know why the authors failed to adjust for this important confounders to put their results into proper perspective. Any explanation for this?
Reviewer 2 Report
Submission of papers is permitted. I think the English expression should be a little softer.Reviewer 3 Report
With real interest, I read the manuscript ijerph-1471868.
Comments:
1. Although the data presented in this manuscript stimulate some controverises or maybe becase of that, they are really interesting. Thus, they deserve tob e presented to the scientific community.
2. However, in my view, the Authors should be even more careful in formulating their conclusions. In other words, the manuscript should be toned down even more. There are several reasons for it.
2A. First, the limitations reported by the Authors themselves, especially using questionnaires (btw. What do you mean “ on CAREGIVERS’ self-report“?).
2B. Second, even though the Authors highlight that some studies by the others yielded nonsignifican data (lines 39-44), their own data seem to be far from significance, at least based on what is presented in Tables 1 and 2. And the single significances present there would probably disappear after correction for multiple testing. Please, add tot he limitations oft he study.
2C. Lines 108-109. “One explanation for our findings may be that only a subset of families engaging in primary industry can be considered to live a farm life“. Indeed, Japan is overall super-highly indistralized and urbanized country, so that some city-village differences can be small.
2D. The hygiene hypothesis is very complex. Please, add a few sentences on it (PMID: 33815389 and 33796103).
2E. Partly in continuation, as partly noted by the Authors, other factors can contribute. There are some factors with two faces, such as cow’s milk (PMID: 31430905 and 33810380).
2F. The differences observed in prevalnces and incidences are small, maybe except for a subcategory “Forestry“ but in this case the numbers of individuals are small.
3. The role of epigenetics in mediating various environmental influences should be mentioned at least in 1-2 sentences in the context of hygiene hypothesis (PMID: 28322581).